# Confronting Patients’ Refusal to Undergo Treatment: A Cross-Sectional Study of Ethical Attitudes and Intended Behaviour Among Lithuanian Physicians

**DOI:** 10.3390/healthcare13222974

**Published:** 2025-11-19

**Authors:** Asta Čekanauskaitė, Karolina Lukošienė, Jelizaveta Krotova, Džiugilė Kersnauskaitė, Benedikt Bachmetjev, Artur Airapetian, Marija Jakubauskienė

**Affiliations:** Faculty of Medicine, Vilnius University, M.K. Ciurlionio 21, LT-03101 Vilnius, Lithuania

**Keywords:** treatment refusal, informed consent, ethics, medical, paternalism, patient discharge

## Abstract

Introduction: Refusal to undergo treatment, including one of its manifestations, discharge against medical advice, must be informed, just as consent is, which is considered a core ethical principle in contemporary medical ethics. The aim of this study was to explore physicians’ attitudes and intended behaviour toward patients’ refusal of necessary treatment and to identify factors associated with their clinical decisions in such situations. Methods: A cross-sectional anonymous online survey of 393 physicians working in Lithuanian public hospitals was conducted between November 2020 and March 2021 using the secure national platform manoapklausa.lt. A convenience sampling strategy ensured representation of both large university and smaller regional hospitals, and all responses were complete due to mandatory fields in the questionnaire. Results: Most physicians (85%; 95% CI 81.2 to 88.5) have encountered situations of refusal to undergo treatment. Women (*p* < 0.05) and senior physicians (aged 51 years and over) (*p* < 0.05) tend to apply treatment without patients’ consent more often in all clinical cases, especially in surgical ones (VN1 and VN2). Difficulty in the decision-making process was associated with chronic conditions and the influence of individual religious beliefs. Insufficient patient information on the intervention was indicated as the main cause of refusal to undergo treatment (62.9%; 95% CI 58.0 to 67.4). Refusal to undergo treatment was associated with physicians’ concern about the patient (57.5%; 95% CI 52.7 to 62.3) and anxiety (38.9%; 95% CI 34.1 to 43.8). Physicians’ attitudes towards patients’ refusal to undergo treatment reflect paternalistic patterns and are mainly associated with the physician’s older age, gender, and duration of professional experience. Insufficient patient information on the intervention was indicated as the most important factor determining patients’ refusal to undergo treatment. Under Lithuanian law, patients have a clear legal right to refuse treatment, and physicians who proceed without consent may face criminal liability. Conclusions: Our findings show that, despite the legal restrictions, many physicians would still choose to treat against a patient’s will, reflecting a persistent paternalistic attitude even in the presence of clear legal prohibitions.

## 1. Introduction

Informed consent is considered the main ethical principle of contemporary medical ethics. Informed consent to medical interventions, as well as refusal to undergo treatment, must be both informed and freely given. Refusal to undergo treatment is defined as the rejection of recommended medications, diagnostic or curative procedures, preventive health measures, hospitalisation, or other medical procedures [1]. Several terms are used in the literature to describe this phenomenon, including self-discharge, refusal of care, refusal to undergo treatment, uncompleted care, and discharge against medical advice (DAMA) [2,3,4,5]. It was first investigated as an issue in the field of psychiatry [3]. Refusal to undergo treatment is particularly common in emergency departments, intensive care units, and prehospital settings. A considerable number of studies focus on the narrower concept of inpatient treatment refusal, i.e., DAMA [3,6].

Discharge Against Medical Advice (DAMA) presents a significant challenge to effective clinical care. The motivations behind DAMA are often unrelated to medical readiness and may reflect personal, social, or contextual pressures [6]. As such, DAMA should not be viewed merely as a private decision, but as a multidimensional issue with broader implications for clinical practice, institutional responsibility, and ethical reasoning.

Among the different types of treatment refusal, Discharge Against Medical Advice (DAMA) stands out because it interrupts ongoing hospital care at a critical stage, often before treatment is complete. It has been shown to carry serious risks for the patient, including higher chances of complications, readmissions, and even death, and it also creates challenges for the healthcare system (both legally and financially). For these reasons, we chose to focus specifically on DAMA in this study. DAMA is defined as a patient’s decision to leave the hospital before the treating physician recommends discharge [4]. However, the issue has recently gained attention across all fields of medicine due to increased sensitivity to its ethical and legal implications.

Recent studies have documented a rising trend in DAMA prevalence across various countries and healthcare settings [7,8,9,10,11]. The literature reports DAMA prevalence from multiple perspectives, including differences across countries, time periods, and medical specialties. Notably, estimated DAMA incidence ranges widely—from 1% to 20% for general medical admissions, and from 6% to 35% for psychiatric admissions [12,13]. In the USA, DAMA prevalence increased from 0.8% in 1993 to a peak of 1.2% in 2015 [7], with recent research indicating even higher rates of up to 2% of all discharges in the USA and 3% in the UK and an overall upward trend [4,8,9,10]. Additionally, paediatric admissions show significant variation internationally, with reported DAMA rates such as 5.3% in Iran, 2% in Singapore, 6% in Israel, 1.2% in Niger, and 1.5–5.7% in Nigeria [11]. Although these paediatric figures are context-specific, they highlight the broad global variability of DAMA. The refusal of recommended treatment remains a challenge affecting both patients and healthcare providers.

The main ethical dilemma lies in the conflict between two core principles of medical ethics: the physician’s obligation to respect patient autonomy and the duty to act in the best interests of the patient. In clinical settings, patients who refuse physicians’ recommendations are often seen as noncompliant, problematic, and time-consuming to manage, revealing remnants of an outdated paternalistic approach [10,14]. Moreover, refusal to undergo treatment raises both medical and economic concerns. Specifically, DAMA has been associated with an increased risk of readmission (often for the same or a related condition), higher morbidity and mortality rates, and even an increased risk of suicide. This, in turn, leads to a higher readmission rate, longer hospital stays, and increased healthcare costs [15].

There is a lack of data on the ethical implications of treatment refusal, particularly regarding the strategies physicians choose in daily clinical practice when facing situations involving refusal to undergo treatment or discharge against medical advice, and how these experiences affect their emotional well-being. Studies have identified several patient-related characteristics and predictors of DAMA, such as male sex, substance abuse, social obligations (e.g., work, childcare), lack of health insurance, lower income level, limited health literacy, poor physical condition, severity of illness, and religious beliefs [16,17]. Additional reasons for leaving the hospital are typically linked to insufficient communication between doctor and patient, inadequate pain management, long waiting times, the teaching hospital setting, and the type of hospital department, especially emergency and oncology units [4,18,19,20,21,22,23].

Furthermore, refusal to undergo treatment is considered a preventable adverse outcome, with improved physician–patient communication identified as the key preventive measure [4]. While the literature extensively analyses reasons for treatment refusal from the patient’s perspective, there is a lack of data exploring this issue from the physician’s point of view. Our study aims to explore physicians’ attitudes and related intended behaviour towards refusal to undergo treatment, and to assess the factors influencing their clinical decisions through four clinical case vignettes. There is still a lack of empirical data on how physicians’ demographic characteristics, such as age, gender, or work experience relate to their attitudes toward patients’ refusal of treatment. Moreover, little is known about how strongly physicians are influenced by the fear of possible legal consequences when facing such situations. Another important yet unexplored question is what proportion of physicians would decide to proceed with treatment despite a patient’s explicit refusal, even when doing so may contradict legal or ethical standards. Addressing these gaps is essential for understanding the persistence of paternalistic patterns in clinical practice and for informing future medical ethics education and policy development.

## 2. Materials and Methods

### 2.1. Study Design and Setting

We conducted a cross-sectional study of Lithuanian physicians. Invitation to participate in the study was distributed to all public healthcare institutions in the country. A link of an online anonymous questionnaire was distributed via the internal communication channels of healthcare facilities. A convenience sample of participants was utilised in this study. Respondents completed the survey voluntarily and of their own accord, without any form of coercion or incentive. As participation was based on self-selection, the sample reflects the views of those individuals who chose to respond, rather than a systematically randomized population. The invitation message briefly described the aim of the study and invited physicians to complete an anonymous online questionnaire. Access to the internal communication system was granted with the permission of the hospital administration. The study targeted only physicians working in public hospitals across Lithuania. No restrictions were applied regarding the medical department, clinical specialty, or level of care, except for the exclusion of dental care professionals. Data collection was conducted from November 2020 through March 2021. This study was conducted in accordance with the Strengthening the Reporting of Observational Studies in Epidemiology (STROBE) guidelines [24].

### 2.2. Study Sampling

The target population for this study was 10,000 licensed physicians working in public hospitals in Lithuania. Eligibility criteria were an active physician license, current practice in a healthcare facility in Lithuania (dental care specialists were excluded), and provision of informed consent. The sample size was estimated using Epi Info 7 software (CDC, USA, 2016), with a 95% confidence interval (CI) and a 5% margin of error. The required sample size was 370 participants. The final sample consisted of 393 physicians. A clustered convenient sampling technique was applied to enrol the study subjects into the survey ensuring coverage of relevant public sector hospitals across the country. The invitation link was distributed through internal communication systems of both large university hospitals and smaller regional institutions to achieve balanced representation of physicians from various specialties and healthcare levels. Although participation was voluntary, the distribution strategy aimed to capture a diverse and realistic cross-section of the Lithuanian public healthcare system.

### 2.3. Tools

To ensure data completeness and integrity, the online survey platform was configured so that all questions were mandatory and had to be answered before the questionnaire could be submitted. As a result, it was not possible for participants to skip any questions or submit incomplete surveys. No missing or corrupted questionnaires were detected in the final dataset, ensuring the quality and reliability of the collected data. The survey was administered online using the Lithuanian platform manoapklausa.lt, which provides secure data collection and complies with national data protection requirements. The platform allowed anonymous participation, ensuring reliability and confidentiality of responses.

### 2.4. Research Tools

Research instrument was composed of three thematic blocks:social and demographic variables (age, work experience, specialisation, working region (urban/rural).four vignettes of clinical cases of different patients refusing recommended treatment in which physicians’ intended behaviour was analysed.physicians‘ attitudes towards patients’ right to refuse treatment: frequency of encountering refusal to undergo treatment situations in their clinical practice, physicians’ perspective on the main reasons for refusal to undergo treatment, decision-making capacity, and how refusal to undergo treatment affects the emotional state of study’s target population.

Thematic block with four vignettes included the following clinical cases of patients refusing treatment or intervention based on different risk-benefit ratio (Table 1).

In each case, respondents were asked to indicate whether they would perform an intervention without the patient’s consent.

The clinical vignettes used in the questionnaire were designed to reflect ethically complex situations that physicians may encounter in real-life practice involving refusal to undergo treatment. The selection of medical conditions (e.g., surgical emergencies, chronic illness, oncological care, and religious objections) aimed to cover a range of urgency levels and treatment contexts, including both somatic and psychological dimensions. The vignettes were developed with input from clinical experts in surgery, internal medicine, and psychiatry to ensure realism and relevance to clinical practice. Gender of the fictional patients was assigned based on clinical plausibility and narrative clarity rather than a balanced design.

The vignettes were presented in a fixed order for all participants. Randomisation of order was not applied, as the primary objective was to assess physicians’ attitudes in relation to specific clinical scenarios, rather than to compare responses across different sequencing conditions.

The list of factors influencing patients’ refusal of treatment was developed by the expert panel that designed the questionnaire, consisting of specialists in ethics, law, and clinical medicine. Respondents were also given the option to add their own answers in an open-text field; however, none of the additional responses differed substantially from the predefined options or appeared among the most frequently selected factors.

The questionnaire was piloted using a smaller sample (*n* = 118) of physicians working in Vilnius city university hospitals in 2019/2020. Based on the results, conceptual amendments were introduced. Afterwards, the questionnaire was reviewed by a multidisciplinary panel, which included experts with legal, ethical, and clinical backgrounds.

### 2.5. Data Analysis

Microsoft Office Excel and IBM SPSS 21.0 software packages were used for statistical analysis. For normally distributed continuous (quantitative) variables, results are presented as mean and standard deviation (SD); for non-normally distributed variables, the median and interquartile range (IQR) are reported. Categorical (qualitative) variables are presented as numbers and percentages. To compare the distribution of categorical variables across groups, the Chi-square (χ^2^) test was used. Binary logistic regression was applied to investigate the prognostic significance of different factors for clinical decisions. First, univariate logistic regression was performed for each variable (with quantitative variables dichotomized at the median). Due to high multicollinearity indicated by a variance inflation factor (VIF) exceeding five between age and work experience, work experience was excluded from the multivariate analysis. This was the only variable removed on this basis. Subsequently, the multivariate logistic regression model was fitted including the remaining variables.

### 2.6. Ethical Considerations

According to Lithuanian law, this anonymous survey did not require ethical approval because it was not considered a biomedical study. This was ensured by two key conditions:Anonymity—No names, surnames, IP addresses, or any other identifying information were collected. Participants answered independently, and their responses could not be traced back to them.Informed Participation—Participants were fully informed about the purpose and nature of the study, including its sensitive topics. A clear explanation was provided at the beginning, allowing them to decide whether to participate. Filling the survey indicated their consent.

Every effort was made to prevent psychological discomfort, and strict measures were taken to protect anonymity, ensuring participants could share their views freely. The study was conducted in accordance with the principles of the Declaration of Helsinki and the Lithuanian Code of Ethics for Researchers, ensuring respect for participants’ rights, autonomy, and confidentiality.

## 3. Results

### 3.1. General Trends in Physicians’ Attitudes and Decision-Making

A total of 393 physicians working in public sector hospitals in Lithuania participated in the study. The questionnaire was distributed via internal hospital intranet systems, targeting actively practicing physicians across multiple institutions. All respondents completed the required items, and no participants were excluded due to missing data.

Participants’ age ranged from 24 to 75 years, with a mean of 43.94 years (SD = 12.73) (Table 2a). Work experience in the medical field ranged from 0 to 54 years, with an average of 17.41 years (SD = 13.14). For both variables, a median split was used to create two categories: 43 years and younger or 44 and older for age, as well as 15 years or less and 16 years or more for work experience. The sample included both general practitioners and medical specialists (Table 2b). However, detailed information on departmental affiliation (e.g., internal medicine, emergency care, surgery) was not collected, and the sample was not stratified by speciality. Therefore, caution is advised when interpreting findings in relation to clinical disciplines.

Across the four clinical vignettes, the percentage of physicians who indicated they would proceed with treatment despite patient refusal was highest in VN2 (73.0%) and VN1 (64.1%), where the scenarios involved acute or urgent conditions. In contrast, a smaller share of respondents chose to override patient wishes in VN3 (29.5%) and VN4 (52.7%), suggesting greater hesitation in chronic or morally complex cases (Table 3).

### 3.2. The Difficulty of Decision Making

Physicians were asked to assess how difficult it was for them to make a clinical decision in each vignette, regardless of whether they chose to follow or override the patient’s wishes. The highest percentage of respondents (63.9%) rated decision-making as “easy” in VN2, where the patient showed signs of anxiety or irritation related to their condition. In contrast, decision-making was perceived as most difficult in VN3 (chronic illness without psychological instability) and VN4 (involving religious or moral objections), with 51.1% and 49.4% of respondents, respectively, reporting that it was “hard to decide.” We did not observe a significant gender difference in reported previous experience with treatment refusal.

### 3.3. The Relationship Between Decision-Making and Socio-Demographic Factors

Women physicians were more likely to perform an intervention without patients’ consent in all clinical cases; however, this difference was statistically significant only in surgical scenarios (VN1 and VN2, *p* = 0.003 and *p* = 0.033, respectively) (Table 4). Age and work experience also showed a clear association with decision-making: senior physicians were more likely to proceed with treatment without consent, especially in surgical situations. Previous experience with treatment refusal further influenced clinical choices; physicians who had not previously encountered such cases were more inclined to treat without consent (significant in VN1 and VN3, *p* = 0.035 and *p* = 0.041, respectively). In contrast, no statistically significant differences were observed based on medical specialisation or worksite location.

To investigate factors associated with physicians’ intended behaviour across four clinical vignettes (VN1–VN4), we performed univariate and multivariate binary logistic regression analyses. In the univariate analyses (Table 5), physician gender was significantly associated with the likelihood of performing an intervention without patient consent in VN1 and VN2, with confidence intervals excluding 1, indicating a reliable effect. Age was a significant predictor in VN1, VN2, and VN3 (all *p* < 0.001) with confidence intervals well below 1, suggesting that older physicians were more inclined to intervene. Years of clinical experience also showed significant associations in VN1 (*p* < 0.001), VN2 (*p* < 0.001), and VN3 (*p* = 0.002), with confidence intervals indicating a robust influence. Previous experience of treatment refusal was significant in VN1 (*p* = 0.037) and VN3 (*p* = 0.043), with confidence intervals not crossing 1, supporting its role as a predictor. Specialization and worksite location were not significantly associated with intended behaviour in any vignette.

We have also performed multivariate logistic regression analyses (Table 6) including gender, continuous age, specialization, worksite location, and prior experience of treatment refusal. Age was a significant predictor in VN1, VN2, and VN3 (*p* < 0.05). Female gender was significantly associated with higher odds of intervention only in VN1 (*p* = 0.0477). Surgical specialization was significant in VN2 (*p* = 0.0436) but not in other vignettes. Worksite location and prior experience of treatment refusal were not significantly associated with intended behaviours in any vignette.

### 3.4. Physicians’ Opinions on the Reasons Why Patients Refuse Treatment

The most frequently reported reasons (Table 7) for patients’ refusal to undergo treatment were insufficient information or poor understanding of the intervention, followed by low education, dissatisfaction with previous medical care, and a history of mental illness.

The most frequently indicated factors were age and fear of pain, followed by a history of oncological disease and family health problems. Less commonly, physicians mentioned concerns about stigmatisation, substance use, and chronic comorbidities, while economic difficulties and long waiting times were rarely identified.

### 3.5. Motivations Behind Physicians’ Decisions to Perform or Withhold Treatment

Across all four samples (VN1–VN4), the most common reason for performing an intervention (Table 8) was the lack of patient competence to decide, particularly high in VN2 (80.8%), while fear of legal consequences was relatively low except in VN4 (25.1%).

Conversely, when not performing an intervention, the dominant reason was respect for the patient’s decision, notably high in VN3 (91.7%) and VN1 (85.1%). Legal concerns again played only a minor role, peaking modestly in VN2 (13.2%).

## 4. Discussion

Refusal to undergo treatment has been most analysed from the patient’s perspective, whereas the standpoint of physicians has rarely been researched. While data on the incidence of treatment refusal encountered by physicians in clinical practice is limited and may vary depending on legal, cultural, and religious contexts, our study showed a high prevalence of this phenomenon: more than 4 out of 5 physicians had experienced treatment refusal in their clinical practice.

Our research aims to identify the factors influencing physicians’ decision-making in various clinical scenarios involving treatment refusal, depending on different risk–benefit ratios. The study showed that it was easier for physicians to make decisions when the case was not clinically complicated and the risk–benefit ratio was clear (e.g., in cases of appendicitis, where the therapeutic benefit was evident). In such situations, paternalistic attitudes and intended behaviours among physicians were more frequent.

However, despite the clear risk–benefit ratio of proposed surgery or blood transfusion in the case of Jehovah’s Witnesses, physicians considered this scenario to be the most ethically challenging. This could possibly be explained by the high public awareness of the Jehovah’s Witnesses’ position regarding blood transfusion, which is well known among medical practitioners and actively advocated by this religious community. In such cases, the decision appeared to be, in part, pre-determined by “public opinion,” which physicians were reluctant to challenge in clinical practice.

### 4.1. Interpretation of Results in the Context of Previous Research

#### 4.1.1. Factors Related to Physicians’ Intended Behaviour

In our study, treatment without informed patient consent was more prevalent among female physicians in 2 out of 4 clinical cases (50%), specifically in surgical scenarios, where the difference was statistically significant. This may suggest a pattern in certain high-risk or procedural contexts but does not generalize across all situations. This finding contrasts with previous studies reporting that female physicians are more likely to adhere to clinical guidelines and engage in patient-centred communication [25,26]. One possible interpretation is that the observed pattern may not reflect paternalism per se but rather be influenced by other factors—such as heightened risk sensitivity, socialised perceptions of professional responsibility, or a stronger inclination to act protectively in situations perceived as high-stakes. Differences in patient demographics, communication expectations, or perceived vulnerability may also play a role. A more detailed investigation would be required to clarify the underlying mechanisms and to avoid overgeneralisation.

Personal experience was important in the decision-making process. Physicians who had previously encountered treatment refusal were more inclined to be less paternalistic and refrained from treating patients without expressed consent. It could be presumed that having prior personal experience with treatment refusal contributed to a broader perspective, encouraged more consideration of the individual situation, and helped to predict possible health consequences more accurately. As a result, decisions were made more easily and with greater clarity.

#### 4.1.2. Work Experience and Age of the Physician

These two factors appeared to be significant in determining the intended behaviour to apply an intervention without the patient’s consent and were associated with a more paternalistic approach. The same trends are confirmed in other studies [27].

#### 4.1.3. Patient-Related Factors to the Treatment Refusal

Physicians in our study highlighted that insufficient information about the intervention provided to the patient was the most important factor related to treatment refusal. This finding is supported by other studies, which suggest that poor communication is one of the main reasons for requesting early discharge, and that investing time in open, non-judgmental communication and collaborative decision-making with the patient is necessary and can contribute to the prevention of DAMA [4].

While long waiting time for the procedure was considered the least important reason for treatment refusal by our respondents, findings from other studies reflecting the patient’s perspective in emergency departments indicate that long waiting time is the second most common reason for leaving the hospital against medical advice [16]. This difference may be explained by the fact that not all clinical situations presented in our study were set in the context of an emergency department, nor did they emphasize waiting time for intervention.

Other studies also report financial difficulties as a common reason for treatment refusal [22,23,27,28], whereas in our study it was a minor contributor. This may be explained by the universal health coverage in Lithuania and the presence of out-of-pocket payments that help meet healthcare needs [29].

#### 4.1.4. Encounters with Treatment Refusal Could Contribute to Physicians’ Burnout

Refusal to undergo treatment often creates moral distress and ethical tension among physicians, as they must balance professional responsibility with respect for patient autonomy [30,31]. Our findings confirm that such situations evoke negative emotions, including anxiety, guilt, and frustration, which are known precursors of burnout and psychological exhaustion [32,33]. This suggests that repeated exposure to ethically challenging cases, such as treatment refusal, may contribute to long-term emotional strain and reduced well-being among physicians.

#### 4.1.5. Legal and Regulatory Context in Lithuania

Refusal of treatment is recognized as a fundamental patient right protected by Lithuanian law. According to Article 8 of the Law on the Rights of Patients and Compensation for the Damage to Their Health (Lietuvos Respublikos pacientų teisių ir žalos sveikatai atlyginimo įstatymas) [34], medical care may be provided only with the patient’s consent, including for minors aged 16 to 18. Therefore, under Lithuanian legislation, every competent patient has the legal right to refuse treatment, and physicians are obliged to respect this decision, even when the refusal may contradict medical recommendations or professional judgment. In such cases, when a physician decides to proceed with treatment despite a competent patient’s explicit refusal, this may give rise to several legal issues:(A.)First of all, under Article 12(4) of the Law on the Practice of Medicine of the Republic of Lithuania (Lietuvos Respublikos medicinos praktikos įstatymas) [35], a physician’s licence may be revoked for a serious violation of patients’ rights. This represents one of the most likely legal consequences in cases where a doctor provides treatment without the patient’s consent. Even if no criminal offence is pursued, such conduct may lead to disciplinary or administrative sanctions, including loss of the medical licence.(B.)Secondly, Article 6.729 of the Civil Code of the Republic of Lithuania (Lietuvos Respublikos civilinis kodeksas) [36] explicitly prohibits providing medical treatment or health care against a patient’s will, except in cases defined by law. This provision reinforces patient autonomy as a fundamental civil right and extends protection even to minors who are capable of understanding their health condition and proposed treatment. It establishes a clear legal boundary obliging physicians to obtain patient consent before any medical intervention.(C.)In addition, depending on the specific circumstances and legal interpretation, performing treatment without the patient’s consent may also fall under criminal liability. According to the Criminal Code of the Republic of Lithuania [37], such actions can be interpreted either as causing physical pain or minor bodily harm (Article 140) or as unlawful self-redress (Article 294). While the legal outcome would depend on judicial assessment and case context, both provisions imply potential criminal prosecution, fines, or imprisonment, highlighting the possible severity of legal consequences for violating patient autonomy.

It is important to emphasize that the mentioned legal provisions apply only when patients are fully conscious, mentally competent, and capable of understanding the meaning and consequences of their decisions [38]. The right to refuse treatment assumes that a person is acting with a clear mind and free will. Therefore, the main difficulty in such cases often lies not in the law itself or in the physician’s values, but in the question of how to assess a person’s mental state and ability to make rational decisions.

This is especially relevant in the first two situations. The elderly man who has lost the meaning of life and the anxious patient who refuses surgery may both be influenced by emotional distress or possible depression, which could limit their ability to decide freely. The third and fourth cases raise slightly different questions. Here, patients seem mentally stable, but their choices are strongly shaped by personal or religious beliefs. This brings up another important question: where is the boundary between free will and belief-driven decision-making? In practice, these cases show that treatment refusal is not only a legal or ethical matter but also a deeply human and philosophical issue that requires understanding the patient’s inner state, motives, and level of awareness.

It is particularly striking that many physicians, despite the serious potential legal consequences of violating patients’ autonomy, did not seem deterred by them. As our data show, only in the fourth case related to religious refusal did about one quarter of respondents express concern about possible legal repercussions. Overall, this reflects a strongly paternalistic attitude among physicians, persisting even in the presence of clear legal prohibitions.

### 4.2. Broader Ethical, Legal, and Practical Implications

To the best of our knowledge, this is the first empirical study in Lithuania and one of the few in Central and Eastern Europe to examine physicians’ attitudes toward patients’ refusal of treatment. While definitions of treatment refusal vary across social and cultural contexts, the issue remains ethically and clinically significant. Our findings reveal a persistent gap between legal obligations and physicians’ moral reasoning: many would still proceed with treatment despite explicit patient refusal. This underscores the need for continuous education in medical ethics and communication, particularly in settings where paternalistic traditions remain strong. Strengthening ethical training and promoting awareness of patient autonomy may help reduce moral distress, improve doctor–patient collaboration, and guide policymakers in developing clearer guidelines and psychological support systems for healthcare professionals.

Based on the results of this study, there are three key practical implications:Education Changes—Doctors with experience in treatment refusals are more likely to respect patient autonomy. Practical experience is important, and more practical training, including simulated refusal scenarios, could prepare medical students for difficult ethical cases.Better Legislation—The study showed that physicians struggle the most with treatment refusals in ethically sensitive cases. This shows the need for better ethical guidelines to help physicians manage these difficult situations without emotional stress or fear of legal consequences.Psychological Support—Refusal to undergo treatment situations often lead to stress among doctors. Physician burnout is a growing concern and healthcare institutions should improve emotional support systems, such as psychological support to help physicians cope with the challenges in such cases.

Future research should focus on understanding how different cultural, legal, and institutional factors influence physician responses to treatment refusals. Expanding research will help develop targeted interventions to improve ethical decision-making and emotional well-being among healthcare professionals.

## 5. Conclusions

The vast majority of physicians working in public sector hospitals in Lithuania demonstrate paternalistic attitudes towards patients’ refusal to undergo treatment. Physician’s gender and age are the major factors contributing to the decision to proceed with treatment without the patient’s informed consent. Lack of communication and insufficient patient information about the intervention were identified by Lithuanian physicians as the most important factors determining patients’ refusal to undergo treatment.

## 6. Limitations of the Study

This study has several limitations that should be considered when interpreting the findings:Sampling and representativeness. The use of a convenience sample with voluntary participation may have introduced self-selection bias, limiting the generalizability of the results to all physicians in Lithuania.Distribution method. The survey was conducted through hospitals’ internal communication systems, ensuring professional relevance but restricting participation mainly to physicians from public healthcare institutions.Study design. The use of clinical vignettes reflects physicians’ stated intentions rather than actual clinical behaviour, which in real practice may be influenced by patient interaction, time pressure, or institutional context.Questionnaire structure. The fixed order of vignettes may have caused minor cognitive fatigue or order effects, potentially influencing responses to later scenarios.Data integrity. Because participation was fully anonymous, the possibility of duplicate submissions could not be completely excluded, although the risk was likely minimal.

## Figures and Tables

**Table 1 healthcare-13-02974-t001:** Detailed descriptions of clinical cases (vignettes) presented to physicians.

VN1 *	An 85-year-old man, living alone in extreme poverty and suffering from multiple chronic conditions (rheumatoid arthritis, ischemic heart disease, and type 2 diabetes), presents to the hospital with acute gangrenous appendicitis. The patient refuses surgery in writing, stating that he no longer sees the point in living after the recent death of his last close relative.
VN2	A 52-year-old man arrives at the emergency department with perforated appendicitis. He appears restless, refuses to listen to medical staff, and insists on being discharged, claiming that he will undergo surgery abroad. There is no contact with relatives and no available information about his medical history. The patient refuses the surgery in writing.
VN3	A 35-year-old woman diagnosed with aggressive breast cancer is advised to undergo surgery and chemotherapy. Despite receiving detailed information about the treatment plan, prognosis, and risks, as well as psychological counselling, she refuses both surgery and chemotherapy, stating that she does not want any interventions on her body.
VN4	A 26-year-old man is hospitalized with bloody diarrhoea lasting two weeks and is diagnosed with toxic dilatation of the colon requiring urgent colectomy. Before the operation, he declares that he is a Jehovah’s Witness and refuses blood transfusions based on his religious beliefs.

* VN1 (Vignette number 1, etc.)—vignette number abbreviation, also used later in the text.

**Table 2 healthcare-13-02974-t002:** (**a**). General demographic of continuous statistics of the respondents; (**b**)**.** General demographic of categorical statistics of the respondents.

(a)					
Characteristic	Min	Median	Max	Mean	St. deviation
Age (years)	24	43	75	43.94	12.73
Work experience (years)	0	15	54	17.41	13.14
(b)					
Characteristic				Frequency	
Gender	Males			19% (*n* = 73)	
	Females			81% (*n* = 320)	
Specialisation	General practitioners			46.1% (*n* = 181)	
	Doctors specialists			52.9% (*n* = 208)	
Worksite location	Urban			72.8% (*n* = 286)	
	Regional			25.2% (*n* = 99)	
Earlier experience of refusal to undergo treatment	Yes			85.0% (*n* = 334)	
	No			15.0% (*n* = 59)	

**Table 3 healthcare-13-02974-t003:** Physicians’ intended behaviour in different clinical cases.

	VN1		VN2		VN3		VN4	
	% (N)	95% CI	% (N)	95% CI	% (N)	95% CI	% (N)	95% CI
Would apply treatment *	64.1 (252)	59.4–68.9	73.0 (287)	68.6–77.4	29.5 (116)	25.0–34.0	52.7 (207)	47.7–57.6
Would NOT apply treatment *	35.9 (141)	31.1–40.6	27.0 (106)	22.6–31.4	70.5 (277)	66.0–75.0	47.3 (186)	42.4–52.3

* Or perform an intervention.

**Table 4 healthcare-13-02974-t004:** Attitudes and related intended behaviour towards refusal to undergo treatment in different groups of physicians.

Characteristics		VN1	*p* Value	VN2	*p* Value	VN3	*p* Value	VN4	*p* Value
Gender	Males	49.3% (*n* = 36)	*p* = 0.003	63.0% (*n* = 46)	*p* = 0.033	26.0% (*n* = 19)	*p* = 0.469	42.5% (*n* = 31)	*p* = 0.053
	Females	67.5% (*n* = 216)		75.3% (*n* = 241)		30.3% (*n* = 97)		55.0% (*n* = 176)	
Age (years)	43 years or <	50.0% (*n* = 100)	*p* < 0.001	63.5% (*n* = 127)	*p* < 0.001	20.0% (*n* = 40)	*p* = <0.001	49.5% (*n* = 99)	*p* = 0.200
	44 and >	78.8% (*n* = 152)		82.9% (*n* = 160)		39.4% (*n* = 76)		55.9% (*n* = 108)	
Experience (years)	15 years or <	51.5% (*n* = 103)	*p* < 0.001	64.5% (*n* = 129)	*p* < 0.001	22.5% (*n* = 45)	*p* = 0.002	49.0% (*n* = 98)	*p* = 0.138
	16 years or >	77.2% (*n* = 149)		81.9% (*n* = 158)		36.8% (*n* = 71)		56.5% (*n* = 109)	
Specialisation	Gen. pract.	63.5% (*n* = 115)	*p* = 0.856	73.5% (*n* = 133)	*p* = 0.845	28.7% (*n* = 52)	*p* = 0.816	56.9% (*n* = 103)	*p* = 0.173
	Specialists	64.4% (*n* = 134)		72.6% (*n* = 151)		29.8% (*n* = 62)		50.0% (*n* = 104)	
Worksite	Urban	61.5% (*n* = 176)	*p* = 0.069	72.0% (*n* = 206)	*p* = 0.264	28.7% (*n* = 82)	*p* = 0.382	53.8% (*n* = 154)	*p* = 0.820
	Regional	71.7% (*n* = 71)		77.8 (*n* = 77)		33.3 (*n* = 33)		52.5% (*n* = 52)	
Experience of refusal	Yes	62.0% (*n* = 207)	*p* = 0.035	72.2% (*n* = 241)	*p* = 0.354	27.5% (n = 92)	*p* = 0.041	52.4% (*n* = 175)	*p* = 0.794
	No	76.3% (*n* = 45)		78.0% (*n* = 46)		40.7% (*n* = 24)		54.2% (*n* = 32)	

**Table 5 healthcare-13-02974-t005:** Results of univariate logistic regression models predicting physicians’ intended behaviour across four clinical scenarios (VN1–VN4). AIC 1743.

Binary Logistic Regression: Univariate Model								
Independent variables	VN1		VN2		VN3		VN4	
	*p*-value	95% CI	*p*-value	95% CI	*p*-value	95% CI	*p*-value	95% CI
Gender	*p* = 0.004	0.280–0.784	0.034	0.326–0.957	0.469	-	0.053	-
Age (43 years or </44 and >)	*p* < 0.001	0.245–0.604	<0.001	0.173–0.420	<0.001	0.224–0.576	0.200	-
Work experience (15 years or </16 or >)	*p* < 0.001	0.203–0.485	<0.001	0.252–0.642	0.002	0.321–0.776	0.138	-
Specialisation	*p* = 0.856	-	0.845	-	0.816	-	0.173	-
Earlier experience of refusal to undergo treatment	*p* = 0.037	0.268–0.961	0.354	-	0.043	0.313–0.983	0.794	-
Worksite location	*p* = 0.069	-	0.264	-	0.382	-	0.820	-

**Table 6 healthcare-13-02974-t006:** Results of multivariate logistic regression models predicting physicians’ intended behaviour across four clinical scenarios (VN1–VN4). AIC 1743.

Multivariate Regression						
	VN1			VN2		
Variable	Odds Ratio	Standard Error	*p*-value	Odds Ratio	Standard Error	*p*-value
(Intercept)	0.6	0.29	0.0737	1.45	0.29	0.2055
Gender (Female)	1.79	0.29	0.0477	1.41	0.3	0.2562
Age	0.86	0.07	0.012	0.51	0.03	0.0001
Specialization (Surgery)	0.7	0.35	0.3118	0.49	0.35	0.0436
Work Location (Rural)	1.11	0.28	0.704	1.03	0.29	0.9087
Experience of treatment refusal	1.54	0.35	0.2123	1.2	0.36	0.6114
	Value					
Log-likelihood	−228.24			−209.4		
AIC	468.48			432.81		
Pseudo-R2 (McFadden)	0.0887			0.0553		
Observations	382			382		
	VN3			VN4		
(Intercept)	0.2	0.32	0	0.79	0.27	0.3795
Gender (Female)	1.23	0.32	0.5219	1.48	0.28	0.1589
Age	0.93	0.12	0.0002	0.69	0.06	0.1518
Specialization (Surgery)	1.24	0.35	0.5427	0.65	0.32	0.1825
Work Location (Rural)	0.93	0.27	0.7884	0.91	0.25	0.6911
Experience of treatment refusal	1.57	0.31	0.1511	1.07	0.3	0.8161
	Value					
Log-likelihood	−222.3			−259.95		
AIC	456.6			531.89		
Pseudo-R2 (McFadden)	0.0417			0.0139		
Observations	382			382		

**Table 7 healthcare-13-02974-t007:** Factors most influencing a patient’s decision to refuse treatment (as assessed by respondents).

Factors	% (N)	CI 95%
Insufficient patient information on intervention and its comprehension	62.8 (247)	58.0–67.4%
Education	46.6 (183)	41.7–51.5%
Doctor’s behaviour/dissatisfaction with previous medical staff	43.3 (170)	38.4–48.2%
History of mental illness	41.0 (161)	36.2–46.0%
Age	38.7 (152)	33.9–43.6%
Fear of pain	37.7 (148)	32.9–42.6%
History of oncological disease	30.5 (120)	25.9–35.3%
Illnesses/early deaths of other family members	29.3 (115)	24.8–34.1%
Fear of stigmatisation by others	27.2 (107)	22.8–31.9%
Use of psychoactive substances	25.7 (101)	21.5–30.4%
Concurrent non-cancerous chronic diseases (diabetes mellitus, hypertension, etc.)	15.3 (60)	11.8–19.1%
Poor economic situation	13.5 (53)	10.3–17.3%
Long waiting time for the intervention/treatment	3.6 (14)	2.0–5.9%

**Table 8 healthcare-13-02974-t008:** Reasons for different physicians’ clinical decisions in situations of refusal to undergo treatment.

	**VN1, *n* = 252**		**VN2, *n* = 287**		**VN3, *n* = 116**		**VN4, *n* = 207**	
Reasons for performing an intervention	(%)	95% CI	(%)	95% CI	(%)	95% CI	(%)	95% CI
The lack of patient’s competence for the decision	54.0	48–60	80.8	76–85	41.4	32–50	44.4	38–51
Fear of legal consequences and facing guilty conscience	17.5	13–22	4.5	2–7	21.6	14–29	25.1	19–31
Other reasons	28.6	20–37	14.6	5–11	37.1	28–46	28.5	22–35
	**VN1, *n* = 141**		**VN2, *n* = 106**		**VN3, *n* = 277**		**VN4, *n* = 186**	
Reasons for NOT performing an intervention	(%)	95% CI	(%)	95% CI	(%)	95% CI	(%)	95% CI
Respect for the patient’s decision	85.1	79–91	68.9	60–78	91.7	88–95	83.3	77–88
Fear of legal consequences and facing guilty conscience	5.0	1–9	13.2	7–20	2.5	1–4	7.5	4–11
Other reasons	9.9	5–15	17.9	11–25	5.8	3–9	9.1	3–5

## Data Availability

The dataset supporting the conclusions of this publication is proprietary and will not be publicly shared. However, additional information regarding the study methodology and analysis can be provided upon reasonable request to the corresponding author.

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
