# Peer review of "Confronting Patients’ Refusal to Undergo Treatment: A Cross-Sectional Study of Ethical Attitudes and Intended Behaviour Among Lithuanian Physicians"

_healthcare, 2025, doi:10.3390/healthcare13222974_

Round 1

Reviewer 1 Report

Comments and Suggestions for Authors

The submitted manuscript consists of 15 pages, including 8 tables and the list of  33 literature references in total. The manuscript based on own original material explores the problem of patients' refusal to submit to necessary treatment as ethical dilemma of physicians in Lithuania. As such, the topic presents as both current and important, thus likely to rise the interest in the Readers of the Journal. The title is misleading as it suggests in the first place that the problem is refusal by the doctors to treat patients and not, as it is in fact, physicians' attitudes towards patients' refusal to be treated even if such a treatment is objectively required, with its extreme variant of discharge against medical advice DAMA, thus the title is not fully relevant to the contents of the main text of the article. The title shall be modified, e.c. "Ethical burden of physicians confronting patients' refusal to submit to necessary medical treatment: Cross-sectional survey of attitudes and related intended behavior among Lithuanian physicians".  The English quality is acceptable, the structure of the text is clear enough - however, the phrase "refusal of treatment" shall be changed into "refusal to submit to treatment/refusal to get treated/refusal to undergo treatment" for much better clarity. The abstract mirrors both the structure and key contents of the main text - the word "introduction" in the very beginning of the abstract text seems to be a obsolete leftover and shall be removed. Abbreviations were explained while used for the first time. The Introduction provides some background information but lacks the crucial legal context. All the cases VN1-VN4 were legally the cases where going forward with treatment would constitute overt violation by the physician of clearly and openly expressed conscious patients' right to object the treatment, guaranteed by law, so in fact physicians must give up treatment, no matter how they feel about it and there is no field for any maneuver - at least in Poland. In this context, asking physicians whether they "would perform an intervention to the patient without one’s consent" (line 151) has not much sense as it translates for the physician into a lost case in both criminal and civil trial - that is obviously avoided as much as possible in practice. Considering the above, the Authors may provide in the article some legal background in Lithuania relevant to the patients' rights to consent/object to treatment, so the Readers can relate it to the range of free decisions for physicians in the country, which may differ significantly from that they know from their own country. In this context the results provided by the Authors are especially appealing - as in any of the presented 4 cases more than 1/3 of the participating Lithuanian physicians would violate the rules that are practically insurmountable in e.c. Poland! Furthermore, the factors influencing the physicians' decisions that were investigated by the Authors (line 280) in Polish reality would be irrelevant, with the exception of factors objectively limiting the patients' legal ability to make informed decisions, like being unconscious, under psychotropic substances influence, in the state of acute psychosis or in deep dementia - thus justifying making the decision on treatment the physicians' call. To the contrast, the Authors' results point at minor role of legal factors in Lithuanian physicians' decisions on the investigated matters (line 295), which even more vividly demands for describing in the article the local legal background, including the basis and severity of sanctions threatening the physicians violating the patients' autonomy by, as the Authors stated in the Conclusion, "paternalistic attitudes towards patients' refusal of treatment" presented by the "vast majority of physicians"(line 419). The literature references are numerous and reasonably recent, they shall be extended with references to legal acts defining the physicians' duties in the context of patients' rights and responsibilities in case of their violations, respective to the topic of the work.

It needs to be stressed that the article is of vital value to the discussion about the real extent of the patients' rights and their practical introduction, paving the way to better regulate these issues in the Lithuanian law, thus it is strongly recommended to publish it. However, it is also strongly recommended that the Authors add detailed description of Lithuanian legal ramifications of the topic to their article, as only this will allow for full and in-depth interpretation of the presented results. 

Line 454: remove orphan "1"

Table 1 lacks lines separating cases VN2-VN4 from each other, they shall be added.

Comments on the Quality of English Language

The English language quality is acceptable.

Author Response

I greatly appreciate the time and effort you have dedicated to reviewing our manuscript. Below, you will find comprehensive responses to each of your comments. Additionally, in the revised manuscript that we will re-submit, all changes and corrections are clearly marked using different colors for each reviewer for your convenience. All changes made in response to your comments will be highlighted in blue for easy identification and reference. As we are also addressing the comments of another reviewer, we will indicate the exact line numbers of the revisions, including those where your comments overlap with those of the other reviewer.

Comment 1: The title is misleading as it suggests in the first place that the problem is refusal by the doctors to treat patients and not, as it is in fact, physicians' attitudes towards patients' refusal to be treated even if such a treatment is objectively required, with its extreme variant of discharge against medical advice DAMA, thus the title is not fully relevant to the contents of the main text of the article. The title shall be modified, e.c. "Ethical burden of physicians confronting patients' refusal to submit to necessary medical treatment: Cross-sectional survey of attitudes and related intended behavior among Lithuanian physicians".  

Response: We fully agree that the initial title could be misleading and have revised it. I hope this version emphasizes physicians’ ethical response to patients’ refusal of treatment while keeping the title concise and stylistically appropriate for the journal.

Comment 2: The English quality is acceptable, the structure of the text is clear enough - however, the phrase "refusal of treatment" shall be changed into "refusal to submit to treatment/refusal to get treated/refusal to undergo treatment" for much better clarity.

Response: We have revised the formulation accordingly and now use the phrase “refusal to undergo treatment” throughout the manuscript to ensure greater clarity and precision.

Comment 3: The abstract mirrors both the structure and key contents of the main text - the word "introduction" in the very beginning of the abstract text seems to be a obsolete leftover and shall be removed. Abbreviations were explained while used for the first time. The Introduction provides some background information but lacks the crucial legal context. All the cases VN1-VN4 were legally the cases where going forward with treatment would constitute overt violation by the physician of clearly and openly expressed conscious patients' right to object the treatment, guaranteed by law, so in fact physicians must give up treatment, no matter how they feel about it and there is no field for any maneuver - at least in Poland. In this context, asking physicians whether they "would perform an intervention to the patient without one’s consent" (line 151) has not much sense as it translates for the physician into a lost case in both criminal and civil trial - that is obviously avoided as much as possible in practice. Considering the above, the Authors may provide in the article some legal background in Lithuania relevant to the patients' rights to consent/object to treatment, so the Readers can relate it to the range of free decisions for physicians in the country, which may differ significantly from that they know from their own country. In this context the results provided by the Authors are especially appealing - as in any of the presented 4 cases more than 1/3 of the participating Lithuanian physicians would violate the rules that are practically insurmountable in e.c. Poland! Furthermore, the factors influencing the physicians' decisions that were investigated by the Authors (line 280) in Polish reality would be irrelevant, with the exception of factors objectively limiting the patients' legal ability to make informed decisions, like being unconscious, under psychotropic substances influence, in the state of acute psychosis or in deep dementia - thus justifying making the decision on treatment the physicians' call. To the contrast, the Authors' results point at minor role of legal factors in Lithuanian physicians' decisions on the investigated matters (line 295), which even more vividly demands for describing in the article the local legal background, including the basis and severity of sanctions threatening the physicians violating the patients' autonomy by, as the Authors stated in the Conclusion, "paternalistic attitudes towards patients' refusal of treatment" presented by the "vast majority of physicians"(line 419). The literature references are numerous and reasonably recent, they shall be extended with references to legal acts defining the physicians' duties in the context of patients' rights and responsibilities in case of their violations, respective to the topic of the work.

Response: Thank you for this thoughtful and valuable comment. Under Lithuanian law, proceeding with treatment against a competent patient’s explicit refusal is a legal violation. We added a section discussing the national legal background, noting that such actions could, in theory, expose physicians to criminal prosecution or even imprisonment under the Lithuanian Criminal Code (we have cited the articles of the Criminal Code). We also clarified that these laws apply only when the patient is fully conscious and mentally competent to make an informed decision. In this context, we expanded the discussion to include a philosophical perspective, addressing the complexity of determining when a person is truly capable of rational choice, especially in cases where emotional distress, depression, or strong religious beliefs may blur the boundaries of free will and decision-making. We have also added that despite serious potential legal consequences, many physicians in our study did not appear deterred by them, which highlights the persistence of paternalistic attitudes even under clear legal prohibitions.

Comment 7: It needs to be stressed that the article is of vital value to the discussion about the real extent of the patients' rights and their practical introduction, paving the way to better regulate these issues in the Lithuanian law, thus it is strongly recommended to publish it. However, it is also strongly recommended that the Authors add detailed description of Lithuanian legal ramifications of the topic to their article, as only this will allow for full and in-depth interpretation of the presented results. 

Response: We hope that the added legal context provides a more comprehensive understanding of the issue.

Comment 8: Line 454: remove orphan "1"

Response: We carefully reviewed the manuscript but were unable to identify the orphan “1” mentioned in line 454. We would kindly appreciate a more specific indication of its location to ensure precise correction.

Comment 9: Table 1 lacks lines separating cases VN2-VN4 from each other, they shall be added.

Response: We have carefully added lines to the table to improve readability and visual clarity.

Reviewer 2 Report

Comments and Suggestions for Authors

                                                 The title should be more precise and clear.

The purpose, time, and location  should be mentioned in the title.

Women (p=0.003 and p=0.033 in VN1 and VN2, respectively) , this is confusing  also line 20-21, its confusing.

In the abstract, the objective should be stated.

The methodology should be more complete, including sampling, tools, and analysis.

Check keywords based on the mesh.

In the introduction, sentences should have references, for example lines 43 to 50.

The necessity of doing the work and the gap should be stated more precisely.

Why A convenience sample?? Are the results generalizable and valid?

Data collection was conducted from November 2020 through March 2021, the data out of data!!

The details of the sample size calculation are not clear. what was the standard deviation?

Was the questionnaire valid and reliable?

No results of the validity and reliability of the questionnaire are observed, so the instrument is not appropriate, and as a result, the results are not reliable.

The  ethical code  is not mentioned.

In Table 6, not all items are required.

For tables, not all results need to be mentioned in the text.

In Table 7, the basis for these questions is not clear.

In the discussion, many items are the researcher's opinion, which is not appropriate and should have appropriate references.

4.4 to 4.6 are vague and unclear.

The limitations are long and incomprehensible.

Given the type of sampling, generalization of results should be done with caution.

Author Response

I greatly appreciate the time and effort you have dedicated to reviewing our manuscript. Below, you will find comprehensive responses to each of your comments. Additionally, in the revised manuscript that we will re-submit, all changes and corrections are clearly marked using different colors for each reviewer for your convenience. All changes made in response to your comments will be highlighted in green for easy identification and reference. As we are also addressing the comments of another reviewer, we will indicate the exact line numbers of the revisions.

Comment 1:  purpose, time, and location should be mentioned in the title.

Response: The revision of the title was also made based on another reviewer’s comment. Following their suggestion, we modified the title to “Confronting Patients’ Refusal of Necessary Treatment: Ethical Attitudes and Intended Behaviour among Lithuanian Physicians”, emphasizing the ethical aspect of the issue and specifying the study location. You have recommended including the study period in the title; After reviewing similar publications, we found that years are rarely specified in article titles. We would appreciate any suggestion on how this could be incorporated in a concise and stylistically appropriate way, if You still considers it necessary. (lines 1-4)

Comment 2: Women (p=0.003 and p=0.033 in VN1 and VN2, respectively) , this is confusing  also line 20-21, its confusing.

Response: We simplified this sentence and replaced the exact values with a general statement indicating that the results were statistically significant (p < 0.05). (lines 22-23)

Comment 3: In the abstract, the objective should be stated.

Response: We have added a clear statement of the study objective in the Abstract. (lines 15-17)

Comment 4: The methodology should be more complete, including sampling, tools, and analysis.

Response: We have expanded the Methodology section (lines 140-145; 152-155) and added a clearer summary of the study design and procedures in the Abstract to improve overall transparency and completeness (lines 17-21).

Comment 5: Check keywords based on the mesh.

Response: We have reviewed and updated the keywords to ensure consistency with official MeSH terminology. (line 37)

Comment 6: In the introduction, sentences should have references, for example lines 43 to 50.

Response: One irrelevant sentence was removed, and the reference was corrected to align with the previously cited sources, which are now properly indicated in the revised version. (line 53)

Comment 7: The necessity of doing the work and the gap should be stated more precisely.

Response: We have clarified the research gap in the Introduction by specifying the lack of data on demographic correlations, physicians’ fear of legal consequences, and the proportion of doctors willing to treat against a patient’s explicit refusal. (lines 105-113)

Comment 8: Why A convenience sample?? Are the results generalizable and valid?

Response: We used a convenience sample because participation was voluntary and based on physicians’ willingness to respond to the online invitation distributed across hospitals nationwide. While this limits full generalizability, the sample included physicians from various regions, specialties, and hospital levels, providing a diverse and realistic reflection of the Lithuanian public healthcare system. This was now clearly mentioned in the limitations' section. (lines 484-500)

Comment 9: Data collection was conducted from November 2020 through March 2021, the data out of data!!

Response: Although the data were collected in 2020–2021, the ethical and legal framework in Lithuania has not changed since then, and the findings remain relevant for understanding physicians’ attitudes toward patient autonomy. The study continues to provide valuable insight into persistent patterns of paternalism that are unlikely to have changed substantially over this short period.

Comment 10: The details of the sample size calculation are not clear. what was the standard deviation?

Response: The sample size calculation was based on proportions rather than continuous variables; therefore, standard deviation is not applicable. The expected proportion (p) was set at 0.5 with a 95% confidence level and a 5% margin of error, providing a conservative estimate and sufficient statistical power.

Comment 11: Was the questionnaire valid and reliable? No results of the validity and reliability of the questionnaire are

observed, so the instrument is not appropriate, and as a result, the results are not reliable.

Response: Thank you for the comment. The questionnaire’s validity, ensured through expert review by a multidisciplinary panel in ethics, law, and clinical medicine, as well as the wide distribution across large and small hospitals, is described in the text. (lines 175-187)

Comment 12: The  ethical code  is not mentioned.

Response: We have now specified that the study was conducted in accordance with the principles outlined in the Declaration of Helsinki and followed the Lithuanian Code of Ethics for Researchers to ensure compliance with ethical standards. (lines 223-227)

Comment 13: In Table 6, not all items are required.

Response: Thank you for the comment. We would appreciate clarification regarding which specific items in Table 6 you consider unnecessary, so that we can adjust the table accordingly without removing potentially relevant information.

Comment 14: For tables, not all results need to be mentioned in the text.

Response:  We have revised the manuscript to avoid repeating all table results in the text and now summarize only the most relevant findings to improve clarity and readability. We have removed some numerical details and redundant results (lines 279-280; 303), while other parts were rephrased (307-309; 312-315).

Comment 15: In Table 7, the basis for these questions is not clear.

Response: We have clarified in the Methods section that the listed factors were developed by the expert panel that created the questionnaire. Respondents were also able to provide their own answers in an open-text field, but none of the additional responses differed significantly from the predefined options or ranked among the most common choices. (lines 188-192)

Comment 16: In the discussion, many items are the researcher's opinion, which is not appropriate and should have appropriate references.

Response: To improve clarity and reduce the amount of subjective interpretation, we have restructured the Discussion section into two parts: 4.1 Interpretation of Results in the Context of Previous Research and 4.2 Broader Ethical, Legal, and Practical Implications. This structure helps distinguish evidence-based analysis from authors’ reflections. We have also restructured the 4.2 , minimizing it to maintain objectivity. Additionally, we strengthened the legal context by adding relevant references to Lithuanian legislation and legal codes concerning patients’ rights and physicians’ liability.

Comment 17: 4.4 to 4.6 are vague and unclear.

Response: We have shortened and merged Sections 4.4 to 4.6 into a new subsection titled 4.2, making it more structured, concise, and clearly connected to the study results. (lines 448-459)

Comment 18: The limitations are long and incomprehensible. Given the type of sampling, generalization of results should be done with caution.

Response: We have revised the Limitations section into a clear bullet-point format, emphasizing issues of sampling accuracy, generalizability, study design, and data integrity to improve readability and transparency. (lines 484-500)

Round 2

Reviewer 2 Report

Comments and Suggestions for Authors

-